# Ileum Proteomics Identifies Distinct Pathways Associated with Different Dietary Doses of Copper–Fructose Interactions: Implications for the Gut–Liver Axis and MASLD

**DOI:** 10.3390/nu16132083

**Published:** 2024-06-29

**Authors:** Manman Xu, Ming Li, Frederick Benz, Michael Merchant, Craig J. McClain, Ming Song

**Affiliations:** 1Department of Medicine, Division of Gastroenterology, Hepatology and Nutrition, University of Louisville School of Medicine, Louisville, KY 40202, USA; xmm1903@ccmu.edu.cn (M.X.); craig.mcclain@louisville.edu (C.J.M.); 2Department of Medicine, Division of Nephrology and Hypertension, University of Louisville School of Medicine, Louisville, KY 40202, USA; ming.li@louisville.edu (M.L.); michael.merchant@louisville.edu (M.M.); 3Hepatobiology & Toxicology Center, University of Louisville School of Medicine, Louisville, KY 40202, USA; 4Department of Pharmacology and Toxicology, University of Louisville School of Medicine, Louisville, KY 40202, USA; frederick.benz@louisville.edu; 5University of Louisville Alcohol Research Center, University of Louisville School of Medicine, Louisville, KY 40202, USA; 6Robley Rex Louisville VAMC, Louisville, KY 40206, USA

**Keywords:** copper, fructose, proteomics, GO, KEGG, gut barrier function, MASLD

## Abstract

The interactions of different dietary doses of copper with fructose contribute to the development of metabolic dysfunction-associated steatotic liver disease (MASLD) via the gut–liver axis. The underlying mechanisms remain elusive. The aim of this study was to identify the specific pathways leading to gut barrier dysfunction in the ileum using a proteomics approach in a rat model. Male weanling Sprague Dawley rats were fed diets with adequate copper (CuA), marginal copper (CuM), or supplemented copper (CuS) in the absence or presence of fructose supplementation (CuAF, CuMF, and CuSF) for 4 weeks. Ileum protein was extracted and analyzed with an LC-MS. A total of 2847 differentially expressed proteins (DEPs) were identified and submitted to functional enrichment analysis. As a result, the ileum proteome and signaling pathways that were differentially altered were revealed. Of note, the CuAF is characterized by the enrichment of oxidative phosphorylation and ribosome as analyzed with the KEGG; the CuMF is characterized by an enriched arachidonic acid metabolism pathway; and focal adhesion, the regulation of the actin cytoskeleton, and tight junction were significantly enriched by the CuSF. In conclusion, our proteomics analysis identified the specific pathways in the ileum related to the different dietary doses of copper–fructose interactions, suggesting that distinct mechanisms in the gut are involved in the development of MASLD.

## 1. Introduction

Copper is an essential trace element serving as a cofactor or dynamic signaling molecule for many biological processes, including mitochondrial respiration, antioxidant defense, neurotransmitter and neuropeptide synthesis, lipolysis, wound healing, and tumor metastasis and growth [1,2,3,4,5]. Cellular copper levels are tightly regulated by the interplay of import, export, and utilization through an array of copper-dependent proteins and membrane transporters, thus maintaining the cellular copper levels within a narrow range. Either copper deficiency or overload owing to genetic or environmental factors disturbs copper homeostasis, leading to a variety of human diseases, including Menkes disease, Wilson disease, and neurodegenerative diseases [6]. On the other hand, copper depletion or restriction are promising strategies in the treatment of cancers due to the essential role of copper in the promotion of cell proliferation, angiogenesis, and metastasis [7,8,9].

Copper is primarily stored in the liver [6]. Both copper overload and deficiency can lead to hepatic steatosis [10,11,12,13,14]. Importantly, metabolic dysfunction-associated steatotic liver disease (MASLD) affects more than 25% of the population in the United States and worldwide [15]. While Wilson disease is a rare disease leading to MASLD as a result of hepatic copper overload [14], low copper bioavailability is a more common etiological factor observed in MASLD patients and has been relatively well defined [10,11,16]. The underlying mechanisms by which disrupted copper homeostasis promotes hepatic steatosis likely contribute to impaired hepatic mitochondrial biogenesis, fatty acid oxidation, and disturbed iron metabolism [10,11,13,17]. Moreover, a variety of manifestations associated with copper deficiency, including hepatic fat accumulation, are exacerbated by dietary fructose intake, highlighting the critical role of the copper–fructose interaction in the development of MASLD and the metabolic syndrome [18,19,20]. Of note, our previous studies demonstrated that not only dietary low-copper but also high-copper interacts with dietary fructose and promotes the development of hepatic steatosis [18,21,22]. Moreover, we found that both low-copper and high-copper diets, in combination with dietary high-fructose, resulted in an impaired gut barrier function and increased gut permeability [23]. Despite the fact that both dietary low- and high-copper/high-fructose feeding led to gut barrier dysfunction, they altered the gut microbial activity differentially [21,23], suggesting that distinct mechanisms might be involved in the associated impaired gut barrier function. In support of this, several studies showed that dietary copper shapes the morphology of the intestine in a dose-dependent manner [24,25,26].

Given that the intracellular copper level may affect cellular functions through the copper regulatory protein machinery and these effects might be cell type-specific [27], the aim of this current study was to identify specific signaling pathways in the ileum leading to an impaired gut barrier function by dietary low- and high-copper/high-fructose interactions using a proteomics approach. We expect the proteomics data will provide novel insights and generate hypotheses for future mechanistic study.

## 2. Materials and Methods

### 2.1. Animal Experiments

Male weanling Sprague Dawley rats (35–45 g) from the Harlan Laboratories (Indianapolis, IN, USA) were fed a purified AIN-76 diet ad libitum with a defined copper content in the form of cupric carbonate (with the calories comprised 69% carbohydrate, 12% fat, and 19% protein). The rats received 6.0 ppm (CuA, Cat# 115591, Dyets Inc., Bethlehem, PA, USA), 1.5 ppm (CuM, Cat# 115590, Dyets Inc., Bethlehem, PA, USA), or 20 ppm (CuS, Cat# 115592, Dyets Inc., Bethlehem, PA, USA) of copper from diets as adequate, marginal, or supplemental doses, respectively, for 4 weeks. The animals had free access to either deionized water or deionized water containing 30% fructose (*w*/*v*) (CuAF, CuMF, and CuSF) at the same time as the different doses of copper diets for 4 weeks. There were a total of 45 rats. Seven or eight of these rats were randomly assigned to each of the six groups (CuA, CuAF, CuM, CuMF, CuS, and CuSF) (Figure 1). The control animals were those fed adequate copper (CuA) with no added fructose. The animals were single-housed in stainless steel cages without bedding in a temperature- and humidity-controlled room with a 12:12 h light–dark cycle. The cage locations were equally distributed between the groups. The fructose-enriched drinking water was changed twice a week. After fasting overnight, all the animals were killed under anesthesia with pentobarbital (a 50 mg/kg I.P. injection). Blood was collected from the inferior vena cava, and the citrated plasma was stored at −80 °C for further analysis. Portions of the distal ileum were snap-frozen with liquid nitrogen and stored at −80 °C for the proteomics analysis. All the studies were approved by the University of Louisville Institutional Animal Care and Use Committee, which is certified by the American Association of Accreditation of Laboratory Animal Care.

### 2.2. Materials

The TMT 10plex and PepMap Trap Cartridge were purchased from Thermo Scientific (Waltham, MA, USA). The 5 μm 3.0 × 150 mm XBridge C18 column was purchased from Waters (Milford, MA, USA). The Ultracel YM-10 filters were purchased from Millipore (Burlington, MA, USA). The UltraMicroSpinC18 column and macroC18 column were purchased from Nest Group (Southboro, MA, USA). The sequencing grade modified trypsin (frozen) was purchased from Promega (Madison, WI, USA). The LC-MS grade solvents were used for the sample processing and LC-MS analysis. The TMT 10 plex involved dissolving 0.8 mg in 41 μL of acetonitrile (ACN). The trypsin included diluting 20 μg of trypsin (~50 μL) with 950 μL of 0.8 M urea.

### 2.3. Trypsin Digestion

A mixture of equal volumes of all the ileum samples was prepared as a reference sample. Aliquots of 25 μg of protein for the 18 ileum samples and 2 aliquots of the reference sample were transferred to the YM10 filters, washed twice with 50 μL of 0.1 M Tris (at a pH of 8.5, centrifuged at 15,000× *g* to remove the solvent), reduced by incubation with 50 μL of 4 mM TCEP/8 M urea at 50 °C for 45 min, and alkylated with 10 mM of iodoacetamide at room temperature for 15 min in the dark. After alkylation, the reagents were removed by centrifugation at 15,000× *g* for 15 min, washed with 50 μL of 0.8 M urea, suspended in 50 μL of 0.8 M urea, mixed with 50 μL of trypsin, and incubated at 37 °C overnight. After digestion, the digests were collected by centrifugation, desalted with UltraMicroSpinC18 columns, and dried by speedvac (ThermoFisher Scientific, Waltham, MA, USA).

### 2.4. TMT 10plex Labeling

The 18 desalted ileum samples were randomly divided into two groups and formed two sets with a desalted reference sample in each group. The samples in each set were dissolved in 50 μL of 100 mM tetramethylammonium bicarbonate (at a pH of 8.5), mixed with 10 μL of one of the TMT 10plex reagents (TMT126 for the reference sample and other reagents being randomly assigned to the ileum samples), incubated at 50 °C for 1 h, mixed with an additional 10 μL of reagents, and incubated for another hour. After incubation, the remaining reagents were scavenged by mixing with 8 μL of 5% NH_2_OH and incubated at room temperature for 15 min. Then, the samples in the same set were combined, acidified with 2.5% formic acid (FA), desalted with macroC18 columns, and dried by speedvac.

### 2.5. High pH Fractionation

The desalted sample was dissolved in 100 μL of ACN/10 mM of NH_4_HCO_2_ (5%/95%) (at a pH of 10), loaded to a XBridge C18 column, and eluted with a binary solvent (solvent A: 5% ACN/95% 10 mM of NH_4_HCO_2_ at a pH of 10; and solvent B: 90% ACN/10% 10 mM of NH_4_HCO_2_ at a pH of 10) gradient at a flow rate of 0.2 mL/min using a Beckman System Gold HPLC. The gradient started at 2% of B for 1 min, then increased from 2% to 5% of B for 4 min, from 5% to 40% of B for 60 min, and from 40% to 98% of B for 10 min for a total of 75 min. The eluates from 5 to 63.5 min were collected as 78 0.75 min fractions and concatenated into 26 fractions (#2–27): the eluates from 0 to 5 min were collected as fraction #1, and the eluates from 63.5 to 75 min were collected as fraction #28. The fractions were dried by speedvac, dissolved in 100 μL of 75% ACN/0.1% FA, filtered through 0.2 μm filters, dried by speedvac, and stored at −20 °C before the LC-MS analysis.

### 2.6. LC-MS

The dried fractions were reconstituted in 15 μL of 5% ACN/0.1% FA, loaded (2 μL) to a PepMap Trap Cartridge by 5 μL/min of 5% ACN/0.1% FA for 5 min, and transferred to an in-house packed C18 column. Ultimate 3000 with a binary solvent (solvent A: 0.1% FA in water; and solvent B: 0.1% FA in ACN) gradient was used for the LC separation. The gradient started with 5% of B at 200 nL/min for 5 min; increased to 27.5% of B at 200 nL/min for 95 min; increased to 90% of B at 300 nL/min for 5 min; maintained at 90% and 300 nL/min for 8 min; decreased to 5% of B at 300 nL/min for 2 min; and decreased to 5% of B at 200 nL/min for 10 min. The eluates from the LC were ionized by a nanospray source, and data dependent acquisition was performed on an Orbitrap QE-HF mass spectrometer with a MS scan followed by up to 12 MS/MS scans. For the MS scan, the mass resolution was 120,000 and the *m/Z* range was 350–1500. For the MS/MS scan, the mass resolution was 60,000, the isolation window was 1.2 *m/Z*, the target AGC was 1 × 10^5^, the fixed low mass was 100 *m/Z*, and the exclusion window was 30 s with a mass tolerance of 5 ppm.

### 2.7. Data Processing

The Proteome Discoverer version 1.4 was used for data processing and both Sequest and Mascot search engines were used. The reviewed rat databases downloaded from NCBI on 091218 were used for processing all the data files. Variable modifications of methionine oxidation, fixed carbamidomethyl on cysteine, and TMT on lysine, as well as peptide N-terminus, were included. The mass tolerances for MS and MS/MS were 20 ppm and 5 mDa, respectively; a maximum of two missed cleavages was allowed; and the peptide matches with minimal length of 6 were reported. The peptide–spectrum matches (PSMs) from a database search were evaluated by the Target Decoy PSM Validator. Only rank one peptides were counted, and the peptides were counted only for the top-scored proteins. High confidence matches (FDR 0.01) were reported and a minimum of two peptide matches were required for the protein identification. For the relative quantification, only unique peptides were used. The area ratios of the samples versus the reference sample were reported as the fold change. The investigators of the proteomics study were blinded to the detailed animal group information.

### 2.8. Bioinformatics Analysis

The list containing the identified differentially expressed proteins (DEPs) was submitted to functional enrichment analyses. The Gene Ontology (GO) and Kyoto Encyclopedia of Genes and Genomes (KEGG) enrichment analyses were performed using Metascape [https://metascape.org/gp/index.html#/main/step1 (accessed on 23 February 2023)]. The significantly altered GO functions and KEGG pathways with a *p* < 0.05 were identified. To further understand the biological significance of the DEPs, the MetaCore Database Analysis Platform [https://portal.genego.com/ (accessed on 23 February 2023)] was used to analyze the canonical pathways through the enrichment function of the Pathway Maps of DEPs. The pathway significance was set to *p* < 0.05.

### 2.9. Western Blot

Equal amounts of protein extracted from the mucosa of the ileum homogenate were loaded and resolved on 4–15% SDS-polyacrylamide gels and transferred to the PVDF membrane (Millipore, Bedford, MA, USA). The membrane was blocked and probed with primary antibody for mitochondrial OXPHOS complexes I to V (Cat#: ab110413, Abcam, Waltham, MA, USA) and β-actin (Cat#: 4970, Cell Signaling, Danvers, MA, USA) overnight at 4 °C, and incubated with the corresponding horseradish peroxidase-conjugated secondary antibody. The protein signals were visualized using the enhanced chemiluminescence system (Amersham Biosciences, Little Chalfont, UK). Ponceau S staining was used as the loading control.

### 2.10. Statistical Analysis

The DEPs that were significantly increased or decreased included the criteria of the average log2 mean normalized fold change > ±0.3 and a *p*-value < 0.05. The rstatix package in R 64 4.0.3 was used to screen the DEPs between the different diet groups, and tests for the normal distribution of data were conducted prior to the analysis. *p* < 0.05 was considered to be significant for the two-sided tests. A principal component analysis (PCA) was performed to segregate the DEPs from different groups. A hierarchical cluster analysis was performed using the heatmap package. All figures and statistical analyses were generated using R 64 4.0.3 [http://www.r-project.org/ (accessed on 8 March 2023)], unless otherwise stated.

## 3. Results

This study is an extension of our previous studies, in which we demonstrated that CuAF, CuMF, and CuSF induce obvious metabolic phenotypes, including hepatic steatosis and/or increased gut permeability compared to the control CuA group in male SD rats [18,21,23]. However, the initial analysis showed that no enrichment was achieved in the CuM and CuS groups. Moreover, no obvious hepatic steatosis was observed in the CuM and CuS groups. Therefore, the proteomics data presented in this study are the CuAF, CuMF, and CuSF groups, compared with the CuA (Figure 1). The ileum samples were analyzed in two separate sets. In total, 3005 protein groups were identified from Set 1 and 3150 protein groups were found from Set 2. Among them, 158 of the protein groups were identified in Set 1 alone, 303 protein groups were identified in Set 2 alone, and 2847 proteins were identified in both sample sets. The protein groups identified from only one set of samples do not have complete quantification data and are not suitable for analysis of the change in the protein amount. Therefore, a total of 2847 proteins were used for the subsequent comparative analysis.

### 3.1. Analysis of Differentially Expressed Proteins (DEPs)

First, the principal component analysis (PCA) was used to segregate the DEPs between different dietary groups. The PCA assigned the different dietary groups into distinct clusters (Figure 2). Then, a Venn diagram showed that a total of 474 proteins were differentially expressed between the CuA and CuAF groups, which was higher than the numbers of proteins that were differentially expressed between the CuA and CuMF or CuA and CuSF groups (53 and 96, respectively). The proportion of non-overlapping DEPs were 90.9%, 39.6%, and 70.8% in the CuAF, CuMF, and CuSF groups, compared to the CuA (Figure 3 and Appendix A), suggesting that the majority of DEPs are distinctly altered by interactions of different dietary doses of copper with fructose and only a minor number of altered DEPs are shared. The volcano plots and heatmaps demonstrate the significantly upregulated and downregulated DEPs from the different comparisons (Figure 4 and Appendix A). While only 15 proteins were significantly upregulated by the CuAF compared to the CuA, with the top five gene names being *H2az1*, *Mtrr*, *Magi1*, *Sipa1l2*, and *Fut11*, a total of 459 proteins were significantly downregulated, with the top five gene names being *Sipa1l1*, *Rpl34*, *Apoh*, *Necap1*, and *Rai14* (Figure 4A,B and Appendix A). Compared to the CuA, 18 proteins were significantly upregulated, and 35 proteins were significantly downregulated in the CuMF group. The gene names of the top five upregulated proteins by the CuMF were *Gch1*, *Cd9*, *Des*, *Ube2l6*, and *Plvap*. The gene names of the top five downregulated proteins by the CuMF were *Lgals3*, *Alpi*, *Cox7c*, *Tubg1*, and *Adam17* (Figure 4C,D and Appendix A). A total of 56 and 40 proteins were significantly upregulated and downregulated, respectively, by the CuSF compared to the CuA. The gene names of the top five upregulated proteins by the CuSF were *Itga1*, *Cd9*, *Fabp4*, *Cavin1*, and *Cavin3*. The gene names of the top five downregulated proteins by the CuSF were *Lgals4*, *H1-0*, *Gsta5*, *Abhd14b*, and *Actn4* (Figure 4E,F and Appendix A). Interestingly, a common upregulated protein, CD9, was found in both the CuMF and CuSF groups compared to the CuA. *Cd9* is a protein-encoding gene that is a member of the transmembrane 4 superfamily also known as the tetraspanin family and is a cell surface glycoprotein involved in a multitude of biological processes such as adhesion, motility, membrane fusion, signaling, and protein trafficking [28]. In addition, CD9 is commonly used as a marker for exosomes as it is contained on their surface [29], suggesting that the CuMF- and CuSF-induced metabolic phenotypes are possibly mediated by exosomes.

Among the top downregulated proteins, galectin-3 and galectin-4, which are encoded by the *Lgals3* gene and the *Lgals4* gene, respectively, were found in the CuMF and CuSF groups, respectively. Galectins are a family of the beta-galactoside-binding protein family that plays an important role in cell–cell adhesion, cell–matrix interactions, macrophage activation, angiogenesis, metastasis, and apoptosis [30]. A common function of galectin-3 and galectin-4 is antimicrobial activity against bacteria, suggesting that both the CuMF and CuSF impair antimicrobial activity, and this likely contributes to the impaired gut barrier function.

### 3.2. Functional Enrichment and Pathway Enrichment Analysis of the DEPs between the CuA and CuAF Groups

To further understand the biological significance exerted by the DEPs, GO and KEGG pathway enrichment analyses were performed. The top 10 GO terms in the biological process (BP), cellular component (CC), and molecular function (MF) are presented, including the peptide metabolic process, peptide biosynthetic process, translation (BP), ribosome, ribosomal subunit, cytosolic ribosome (CC), structural molecule activity, structural constituent of ribosome, and mRNA binding (MF) (Figure 5A). The top 20 pathways identified by the KEGG analysis were categorized and are presented in Figure 5B and Appendix A, including oxidative phosphorylation, ribosome, and coronavirus disease—COVID-19. To validate these findings, we performed a western blot analysis for mitochondrial oxidative phosphorylation complexes in the ileum and found that the protein expression of complex III and complex V were significantly downregulated in the CuAF compared to the CuA (Figure 5D), supporting the concept that oxidative phosphorylation is critically involved in the CuAF-induced signaling alteration. Collectively, oxidative phosphorylation and ribosome are likely pathways induced by CuAF.

To better understand the signaling pathways altered by the CuAF, we performed a pathway enrichment analysis using the MetaCore Database Analysis Platform through the enrichment function of the Pathway Maps. The top 10 pathways are presented in Figure 5C and Appendix A, including signal transduction_HTR2A signaling outside the nervous system, NETosis in SLE, Signal transduction_Muscarinic acetylcholine receptors signaling to second messengers, and others.

### 3.3. Functional Enrichment and Pathway Enrichment Analysis of the DEPs between the CuA and CuMF Groups

The top 10 GO terms by the CuMF in the BP and CC are presented in Figure 6A; however, only five GO terms in the MF were enriched by the CuMF. Among them, the GO terms involved in the BP included the positive regulation of cell migration, positive regulation of cell motility, positive regulation of locomotion, and regulation of leukocyte migration. The GO terms involved in the CC included the perinuclear region of cytoplasm, side of membrane, membrane raft, and membrane microdomain. The top enriched GO terms in the MF included oxidoreductase activity, integrin binding, cell adhesion molecule binding, phosphatase binding, and iron–ion binding. A total of 17 pathways were enriched by the CuMF as shown through the KEGG pathway analysis. Among them, arachidonic acid metabolism was identified to be the top enriched metabolic pathway. In addition, linoleic acid metabolism, oxidative phosphorylation, and ferroptosis were also enriched by the CuMF compared to the CuA (Figure 6B and Appendix A). Among the top 10 enriched pathways identified by MetaCore (Figure 6C and Appendix A), arachidonic acid (AA) metabolism seems to be a common pathway identified by both the KEGG and MetaCore, highlighting a specific role of AA metabolism in CuMF-induced gut barrier function.

### 3.4. Functional Enrichment and Pathway Enrichment Analysis of the DEPs between the CuA and CuSF Groups

The top 10 enriched GO terms by the CuSF are shown in Figure 7A, including actin cytoskeleton organization, actin filament-based process in the BP, focal adhesion, cell–substrate junction in the CC, actin binding, exopeptidase activity, actin filament binding, alpha–actinin binding, actinin binding, and integrin binding in the MF. Consistently, the KEGG analysis identified focal adhesion to be among the top enriched pathways. Strikingly, the regulation of the actin cytoskeleton, adherens junction, and tight junction were also enriched (Figure 7B and Appendix A). Similar results were obtained using MetaCore, such as cell adhesion_integrin-mediated cell adhesion and migration, role of adhesion of SCLC cells in tumor progression, cell adhesion_cell-matrix glycoconjugates, cytoskeleton remodeling_regulation of actin cytoskeleton organization by the kinase effectors of Rho GTPases, cell adhesion_integrin inside-out signaling, etc. (Figure 7C and Appendix A). Of note, most of the enriched GO terms and signaling pathways by the CuSF are directly involved in the gut barrier function.

## 4. Discussion

Our previous studies demonstrated that the interactions of different dietary doses of copper with fructose differentially altered the gut microbiome activity and the tight junction proteins [21,23], suggesting that distinct mechanisms might be involved in the context of the gut barrier function. In this current study, we further identified the specific signaling pathways in the ileum proteome from rats fed with different doses of copper–fructose.

First, the PCA showed that the ileum DEPs cluster differentially in the CuAF, CuMF, and CuSF groups compared to the CuA. A Venn diagram analysis further showed that several DEPs were differentially altered by the CuAF, CuMF, and CuSF, relative to the CuA. Additionally, volcano plots and heatmaps showed that distinct upregulated and downregulated DEPs were associated with the CuAF, CuMF, and CuSF, compared to the CuA. Finally, the GO and KEGG enrichment analysis identified the specific pathways induced by different doses of copper–fructose interactions. Moreover, some of the pathways had common findings in both Metascape and MetaCore, enhancing the reliability of the results. Even though both the CuMF and CuSF lead to impaired gut barrier function, they differentially altered the ileum proteome and signaling pathways.

In this study, CuA was set as the control group. A greater number of altered proteins in the ileum were induced by the CuAF and fewer by the CuSF and CuMF compared to the CuA, suggesting that copper limitation and excess may mitigate fructose-induced differential protein expression. Increasing evidence has demonstrated that fructose activates mTOR signaling [31,32,33,34], which plays a fundamental role in protein synthesis and cell proliferation [35]. We postulate that this effect may be dampened by either copper deficiency or excess. Copper–iron interactions have been well documented. Both copper excess and deficiency may affect iron transport [3,36], leading to iron dyshomeostasis, including anemia [37]. Disturbed iron homeostasis may lead to impaired protein synthesis [38]. Thus, it is possible that copper excess and deficiency may affect protein synthesis and potentially contribute to the lower detection rate of differentially expressed proteins by fructose in the CuMF and CuSF groups. Of note, despite the fact that the total number of DEPs is smaller in the CuMF and CuSF, their identity is largely non-overlapping with the CuAF (53% non-overlapping in the CuMF; 78% in the CuSF), suggesting that copper limitation and excess distinctly alter the ileum proteome.

Among the proteins significantly altered by CuAF, the majority were downregulated, which appears to be the result of the dietary fructose. The KEGG enrichment analysis revealed that oxidative phosphorylation is the most highly enriched metabolic pathway, as induced by dietary fructose under the condition of adequate copper. Since dietary fructose absorption is limited in the upper GI tract, the unabsorbed fructose goes to the distal intestine and is metabolized by gut microbiota, which subsequently leads to gut microbiota dysbiosis and altered gut microbial activity [39]. One of the features is reduced fecal and cecal short-chain fatty acids (SCFAs) [21,40], especially butyrate, which is the major fuel of enterocytes. Gut microbiota-derived butyrate is metabolized by enterocytes via mitochondrial β-oxidation, which consumes oxygen, leading to “physiological hypoxia” and hypoxia inducible factor (HIF) activation, and thus maintains the gut barrier function and host–microbe homeostasis [41]. It is conceivable that reduced fecal SCFAs, likely owing to reduced generation, may lead to metabolic reprogramming in the enterocytes, which dampens the mitochondrial respiration, leading to reduced oxygen consumption and the subsequent disruption of “physiological hypoxia” and the gut barrier function. In support of this, several groups have demonstrated that fructose induced gut barrier dysfunction [42,43].

Arachidonic acid (AA) metabolism was identified to be the top enriched metabolic pathway by the KEGG analysis in the ileum proteome of the CuMF rats compared to the CuA control rats. A similar result was obtained using MetaCore. AA can be obtained exogenously or can be endogenously synthesized from the 18 carbon essential fatty acid (EFA) and linoleic acid (LA) through a series of desaturation and elongation reactions, and it can be further oxidized enzymatically to eicosanoids, which are bioactive signaling lipids, by the cyclooxygenase (COX), lipoxygenase (LOX), and cytochrome P450 (CYP) enzymes, or via non-enzymatic free radical mechanisms [44,45]. In general, AA-derived eicosanoids are thought to be proinflammatory [46]. Moreover, AA is a polyunsaturated fatty acid and is susceptible to lipid peroxidation. Importantly, copper deficiency and fructose ingestion results in decreased antioxidant defense and increased oxidative stress [18]. Previous studies have shown that the AA level was increased in the plasma, liver, and heart of rats fed with a low-copper diet [47,48]. However, how CuMF regulates the AA metabolism in the intestine epithelium remains elusive. Collectively, our data suggest that CuMF-induced gut barrier dysfunction may be mediated through the AA metabolism, possibly owing to the proinflammatory response and oxidative stress-induced cellular injury. However, this hypothesis needs to be tested by future work.

Focal adhesion, the adherens junction, the regulation of the actin cytoskeleton, and the tight junction were the top enriched cellular processes in the ileum proteome of CuSF rats identified by the KEGG analysis. The genes involved are *Itgb1*, *Itga1*, *Parva*, *Actn4*, *Actn1*, *Vcl*, and *Flnc*, which encode proteins, including integrin subunit beta 1, integrin subunit alpha 1, parvin alpha, actinin alpha 4, actinin alpha 1, vinculin, and filamin C, with functional roles involved in cell adhesion, reorganization of the actin cytoskeleton, and cell–cell and cell–matrix junctions [49,50,51]. Consistently, GO terms involved in molecular function, including actin binding, exopeptidase activity, actin filament binding, alpha–actinin binding, actinin binding, integrin binding, and cell adhesion molecule binding are enriched by CuSF. A recent study revealed that copper co-localized with F-actin in dendritic spines, suggesting that copper might regulate the cytoskeleton and directly or indirectly bind to F-actin [52]. Chen and colleagues showed that copper supplementation induced cytoskeleton remodeling through the upregulation of HIF-1α in bone marrow mesenchymal stem cells (BMSCs) [53]. However, how copper regulates the cytoskeleton in the gut epithelial cells remains elusive.

The intestine is exposed to high levels of endogenous and exogenous proteases. Emerging evidence has demonstrated that proteases, such as matrix metalloproteases (MMPs), A disintegrin and metalloproteinases (ADAMs), serine protease, and cysteine protease, play key roles in the regulation of gut permeability via the proteolysis of the cell surface molecules, including adhesion molecules, intercellular junction proteins, and structural molecules. In addition, proteases regulate the activity and availability of cytokines and growth factors, which are also known modulators of intestinal permeability [54]. It is known that copper differentially regulates a variety of protease activities, including cysteine-, aspartic-, metallo-, and serine-proteases [55,56,57,58], which have been demonstrated to be able to degrade tight junction proteins and increase the gut permeability [59,60,61,62]. ADAMs and MMPs are zinc metalloproteinases that catalyze the “ectodomain shedding” of a variety of cell surface proteins including signaling and adhesion molecules. Of note, the ADAMs and MMPs-mediated proteolysis of E-cadherin is regulated by copper [63]. In addition, copper can activate the phosphatidylinositol-3-kinase (PI3K)/Akt pathway [64], which acts upstream of the metalloproteinases [65].

Despite the fact that more proteins were downregulated, the phenotypical alterations, including gut permeability and hepatic fat accumulation, were not prominently altered when comparing the CuAF to the CuMF and CuSF [21,23]. Moreover, the enrichment of pathways was not evident in the CuM or CuS alone. However, significantly enriched pathways were observed in the CuMF and CuSF with the KEGG analysis, suggesting that the copper–fructose interactions are critical to the enrichment of pathways in the ileum and to the development of metabolic phenotypes.

It remains to be determined how the complex interplay between host and microbes alters the intestine signaling pathways and barrier function in the context of different dietary doses of copper–fructose interactions. Our previous study showed that the interactions of different dietary doses of copper with fructose lead to distinct alterations of gut microbial activity. Of note, either a low copper or high copper diet led to *Akkermansia* depletion with a more robust effect in a high copper diet [23]. *Akkermansia muciniphila* is a mucin-degrading bacterium that resides in the mucus layer [66]. The abundance of *Akkermansia muciniphila* was markedly decreased in obese and diabetic mice, which was associated with a reduced thickness of the mucus layer and an impaired gut barrier function. Conversely, the colonization of *Akkermansia muciniphila* restored the mucus layer thickness and gut barrier function, as well as improved the metabolic phenotypes [67]. Consistently, human studies have shown a negative correlation between *Akkermansia muciniphila* abundance and obesity and type 2 diabetes mellitus [68,69]. Given that intestinal mucin is a copper chaperone that captures the copper ion and protects the intestinal cells from copper toxicity [70], it raises a hypothesis that reduced *Akkermansia muciniphila* could decrease the mucus layer thickness and impair the gut barrier function, which potentially could increase the copper exposure/toxicity to the enterocytes. On the other hand, the intracellular copper levels may dose-dependently alter the copper regulatory machinery proteins, which could potentially lead to the activation or inactivation of specific signaling pathways [27].

Given that the pathogenesis of MASLD is complex and highly heterogeneous, precision medicine in the MASLD treatment is challenging. Our study identified distinct signaling pathways in the ileum in a preclinical model of copper homeostasis, which emerged as an important risk factor in MASLD [10,11,13]. Therefore, it provides evidence for potential therapeutic targets using precision medicine.

Several limitations of this study warrant further validation. First, the sample size is small with potential variations, and the comparisons were between two groups. Therefore, the potential risk of a false discovery rate will increase. Although we did some validation studies using western blot analysis, more biological function and mechanistic studies are necessary to corroborate the findings. Second, the focus was solely on male animals. Sex differences remain to be addressed.

## 5. Conclusions

In summary, our proteomics data reveal that distinct signaling pathways in the gut are involved in the different dietary doses of copper–fructose interactions, suggesting that different mechanisms in the gut–liver axis can lead to the development of MASLD. The results provide novel insights into the molecular mechanisms underlying dietary copper–fructose interaction-induced MASLD.

## Figures and Tables

**Figure 1 nutrients-16-02083-f001:**
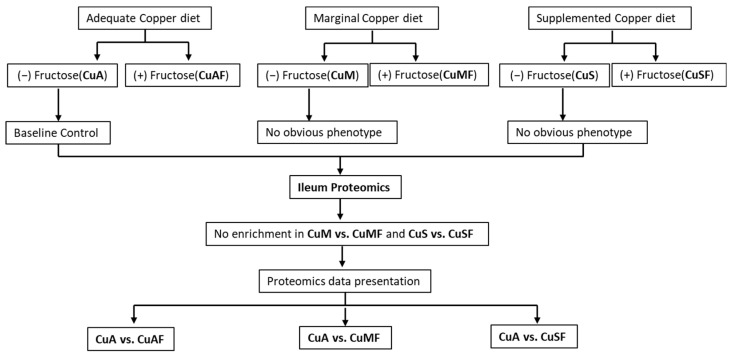
Schematic overview of grouping information and experimental design. CuA, adequate copper; CuAF, adequate copper plus fructose; CuMF, marginal copper plus fructose; and CuSF, supplemented copper plus fructose.

**Figure 2 nutrients-16-02083-f002:**
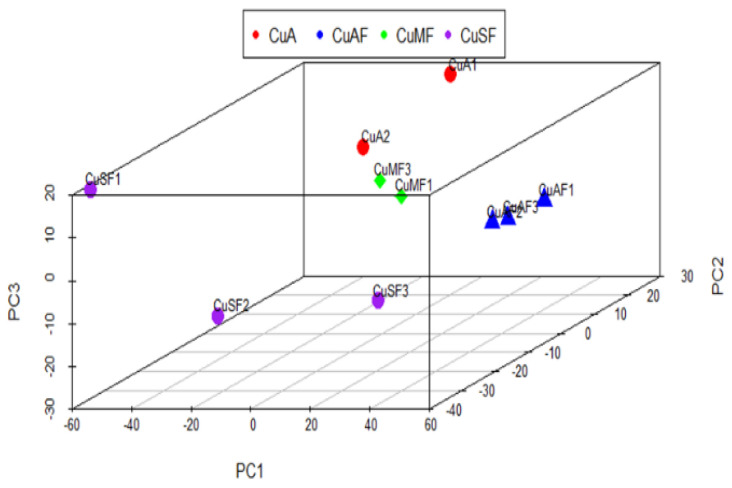
Principal component analysis (PCA) of the proteomics profile differences between different dietary groups: CuA, adequate copper; CuAF, adequate copper plus fructose; CuMF, marginal copper plus fructose; and CuSF, supplemented copper plus fructose.

**Figure 3 nutrients-16-02083-f003:**
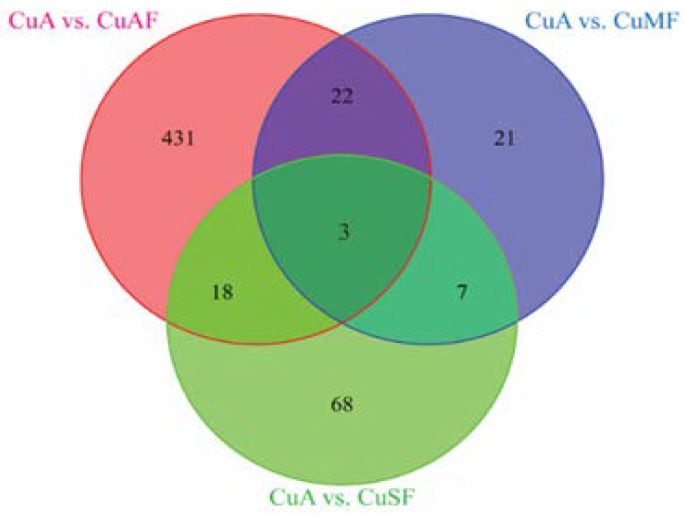
The numbers of the commonly and differentially altered proteins from different groups presented in a Venn diagram: CuA vs. CuAF (pink circle, 474), CuA vs. CuMF (blue circle, 53), and CuA vs. CuSF (green circle, 96). CuA, adequate copper; CuAF, adequate copper plus fructose; CuMF, marginal copper plus fructose; and CuSF, supplemented copper plus fructose.

**Figure 4 nutrients-16-02083-f004:**
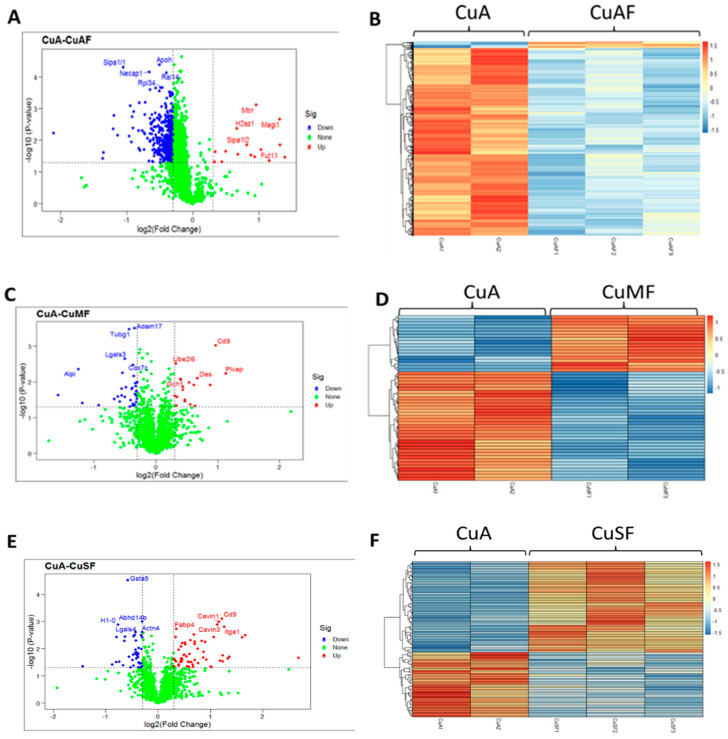
Volcano plots and heatmaps are presented showing the significantly upregulated and downregulated proteins. (**A**,**C**,**E**) Volcano plots from different group comparisons. The blue points denote the significantly downregulated proteins (*p* < 0.05 and log2 fold change < −0.3); the red points denote the significantly upregulated proteins (*p* < 0.05 and log2 fold change > 0.3); and the green points represent the proteins without significant change. (**B**,**D**,**F**) Heatmaps of DEPs from different group comparisons. The sample cluster heatmap analysis was performed with hierarchical clustering using the Pearson correlation method. Dark orange represents the upregulated proteins, while light blue represents the downregulated proteins. (**A**,**B**) CuA vs. CuAF; (**C**,**D**) CuA vs. CuMF; and (**E**,**F**) CuA vs. CuSF. CuA, adequate copper; CuAF, adequate copper plus fructose; CuMF, marginal copper plus fructose; and CuSF, supplemented copper plus fructose.

**Figure 5 nutrients-16-02083-f005:**
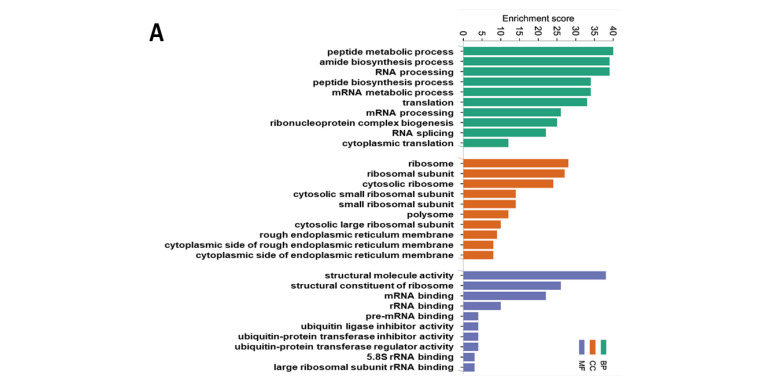
Functional enrichment and pathway enrichment analysis of the DEPs between the CuA and CuAF groups. (**A**) Gene ontology (GO) enrichment analysis; (**B**) top 20 from the Kyoto Encyclopedia of Genes and Genomes (KEGG) pathways analysis results using Metascape; (**C**) top 10 pathways assessed using MetaCore, with the significance of the pathways being shown by the −log(*p*-value); and (**D**) a western blot of the expression of mitochondrial OXPHOS complexes I to V. CuA, adequate copper; CuAF, adequate copper plus fructose; and CI-CV, complex I to complex V.

**Figure 6 nutrients-16-02083-f006:**
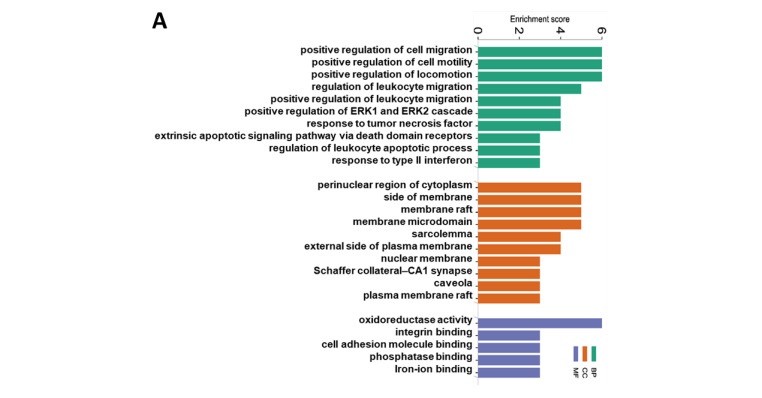
Functional enrichment and pathway enrichment analysis of the DEPs between the CuA and CuMF groups. (**A**) Gene ontology (GO) enrichment analysis; (**B**) the Kyoto Encyclopedia of Genes and Genomes (KEGG) pathways analysis results using Metascape; (**C**) top 10 pathways analyzed using MetaCore with the significance of the pathways being shown by the −log(*p*-value). CuA, adequate copper; and CuMF, marginal copper plus fructose.

**Figure 7 nutrients-16-02083-f007:**
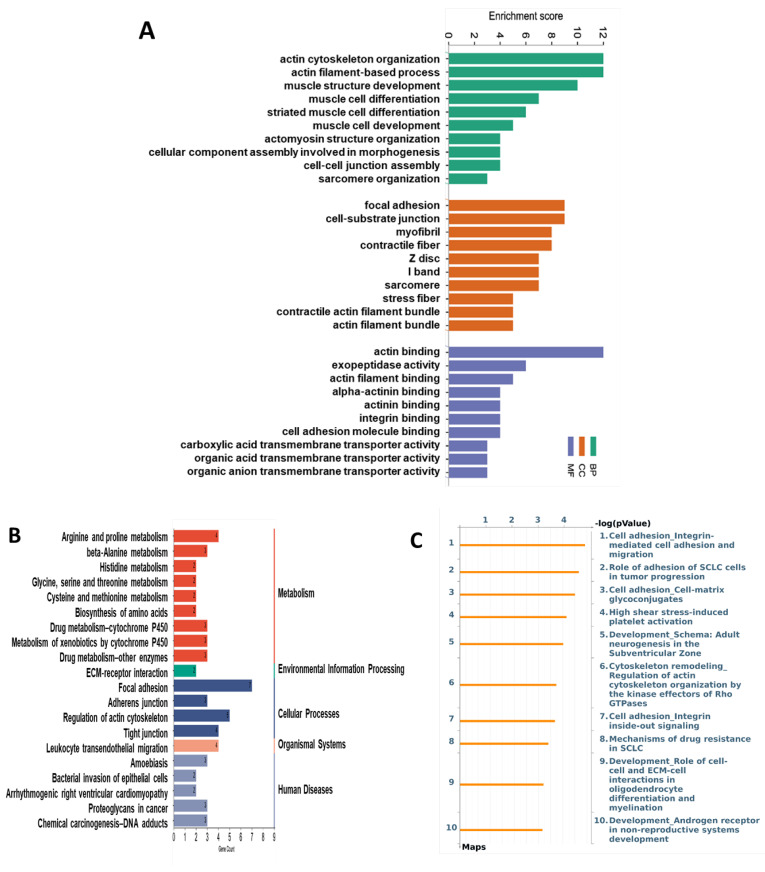
Functional enrichment and pathway enrichment analysis of the DEPs between the CuA and CuSF groups. (**A**) Gene ontology (GO) enrichment analysis; (**B**) top 20 of the Kyoto Encyclopedia of Genes and Genomes (KEGG) pathways analysis results using Metascape; and (**C**) top 10 of pathways analyzed using MetaCore with the significance of the pathways being shown by the −log(*p*-value). CuA, adequate copper; and CuSF, supplemented copper plus fructose.

## Data Availability

All data are contained within this manuscript.

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
