# Peer review of "Ileum Proteomics Identifies Distinct Pathways Associated with Different Dietary Doses of Copper–Fructose Interactions: Implications for the Gut–Liver Axis and MASLD"

_nutrients, 2024, doi:10.3390/nu16132083_

Round 1

Reviewer 1 Report

Comments and Suggestions for Authors

Dear Editor,

I carefully read the manuscript "Ileum Proteomics Identifies Distinct Pathways Associated with Different Dietary Doses of Copper-Fructose Interactions: Implications in the Gut-liver Axis and MASLD".

My comments and suggestions for the authors are the following:

 - The article needs to be formatted following the Instructions for the Authors of the Journal.

 - The limitations of the study should be further and more deeply discussed by the authors.

 - Maybe the authors should enrich the discussion by referring to the clinical implications of their observations.

 - The authors used a Student’s t-test to screen DEPs between the diet groups. By the way, they should have checked for the normal distribution of the variable, before. Was it done?

Reviewer 2 Report

Comments and Suggestions for Authors

GENERAL COMMENTS TO AUTHORS:

The paper by Manman Xu et al. describes the metabolic pathways in ileum associated with the administration of different doses of dietary copper and fructose. Although the work is interesting and fits well within the scope of the journal, there are major concerns that should be addressed, particularly in relation to the experimental design and statistical analysis of the data.

SPECIFIC COMMENTS:

Please find specific comments below:

ABSTRACT: The design of experiment in the abstract is confusing. Please do not describe extensively which groups were compared to which groups (it should be written in the MM section) but briefly mention the methods performed in the paper followed with results and the main conclusion.

INTRO:

Introduction is well written.

MM

Line 82: please unify number terms in percentages or rearrange the sentence (just for example: rats were given standard diet (SD, 11.4% fat, 62,8% carbohydrates, 25,8% proteins; Mucedola, Settimo Milanese, Italy) or any other type of diet.

Line 87: The authors should formulate the structure of the study more clearly, as it is difficult to follow groups and treatment in the present form. How many groups are there in the experiment? Are there groups with different copper treatments and without fructose or do the copper treated groups receive fructose after the copper treatment or at the same time? This should be clearly stated. Please provide a schematic overview of the experimental design.

GENERAL QUESTION: Why did the authors decide to perform the experiment only on male rats? This should be mentioned in the study.

RESULTS:

GENERAL COMMENT: the experimental design is questionable and therefore cannot provide an appropriate conclusion. Therefore, I would like to suggest the authors to revise the experimental design and change it according to the comments below:

Lines 186-189. The statistical problem with the comparisons is that you are comparing the CuAF, CuMF and CuSF groups to a single control group (CuA) without taking into account the possible influence or lack of significance of the CuM and CuS groups.

There is also a problem with multiple comparisons, i.e. if you compare several treatment groups (CuAF, CuMF, CuSF) consecutively with a single control group (CuA), you increase the risk of type I errors (false positive results). Each comparison increases the probability of finding a significant result purely by chance. Instead, use more appropriate statistical methods such as one-way or two-way ANOVA (if applicable) to avoid the likelihood of false significance between groups.

Lines 274-275: Figure 4D Western blot analysis: Please provide the raw photo of the loading control (Ponceau S) as it is difficult to recognise the clear bands on this photo. Also, the loading does not appear to be the same when comparing the membrane to the beta-actin immunoblots. Therefore, it is imperative that the authors provide an appropriate loading control that is clearly visible.

Why do the authors claim that OXPHOS is only affected based on OXPHOS protein expression? To draw a comprehensive and valid conclusion about the functionality of the ETC, the authors should measure functional parameters such as mitochondrial respiration or potential to determine that mitochondrial function was indeed impaired.

Comments on the Quality of English Language

minor editing required.

Round 2

Reviewer 1 Report

Comments and Suggestions for Authors

I carefully read the revised version of the manuscript, that is significantly improved compared to the original version.

Author Response

Response: We thank this reviewer for the help with the improvement of our paper.

Reviewer 2 Report

Comments and Suggestions for Authors

The authors have responded adequately to most of my concerns. Therefore, the paper needs the following minor concerns.

Line 30. delete the extra point.

Line 88. delete part of the sentence “(random numbers generated by Microsoft excel)”

Lines 194 to 200. please delete the extra spaces between words where necessary.

Figure 5 ponceau S is still poorly visible. It would be good to provide a clearer picture of Ponceau
